# DynamicViT: Efficient Vision Transformers with Dynamic Token Sparsification

**Yongming Rao**[1]   **Wenliang Zhao**[1]   **Benlin Liu**[2,3]
**Jiwen Lu**[1*]   **Jie Zhou**[1]   **Cho-Jui Hsieh**[2]

[1] Tsinghua University    [2] UCLA    [3] University of Washington

## Abstract

Attention is sparse in vision transformers. We observe the final prediction in vision transformers is only based on a subset of most informative tokens, which is sufficient for accurate image recognition. Based on this observation, we propose a dynamic token sparsification framework to prune redundant tokens progressively and dynamically based on the input. Specifically, we devise a lightweight prediction module to estimate the importance score of each token given the current features. The module is added to different layers to prune redundant tokens hierarchically. To optimize the prediction module in an end-to-end manner, we propose an attention masking strategy to differentiably prune a token by blocking its interactions with other tokens. Benefiting from the nature of self-attention, the unstructured sparse tokens are still hardware friendly, which makes our framework easy to achieve actual speed-up. By hierarchically pruning 66% of the input tokens, our method greatly reduces $31\% \sim 37\%$ FLOPs and improves the throughput by over 40% while the drop of accuracy is within 0.5% for various vision transformers. Equipped with the dynamic token sparsification framework, DynamicViT models can achieve very competitive complexity/accuracy trade-offs compared to state-of-the-art CNNs and vision transformers on ImageNet. Code is available at https://github.com/raoyongming/DynamicViT.

## 1   Introduction

These years have witnessed the great progress in computer vision brought by the evolution of CNN-type architectures [12, 18]. Some recent works start to replace CNN by using transformer for many vision tasks, like object detection [36, 20] and classification [25]. Just like what has been done to the CNN-type architectures in the past few years, it is also desirable to accelerate the transformer-like models to make them more suitable for real-time applications.

One common practice for the acceleration of CNN-type networks is to prune the filters that are of less importance. The way input is processed by the vision transformer and its variants, *i.e.* splitting the input image into multiple independent patches, provides us another orthogonal way to introduce the sparsity for the acceleration. That is, we can prune the tokens of less importance in the input instance, given the fact that many tokens contribute very little to the final prediction. This is only possible for the transformer-like models where the self-attention module can take the token sequence of variable length as input, and the unstructured pruned input will not affect the self-attention module, while dropping a certain part of the pixels can not really accelerate the convolution operation since the unstructured neighborhood used by convolution would make it difficult to accelerate through parallel computing. Since the hierarchical architecture of CNNs with structural downsampling has improved model efficiency in various vision tasks, we hope to explore the *unstructured* and *data-dependent*

---

*Corresponding author.

35th Conference on Neural Information Processing Systems (NeurIPS 2021).

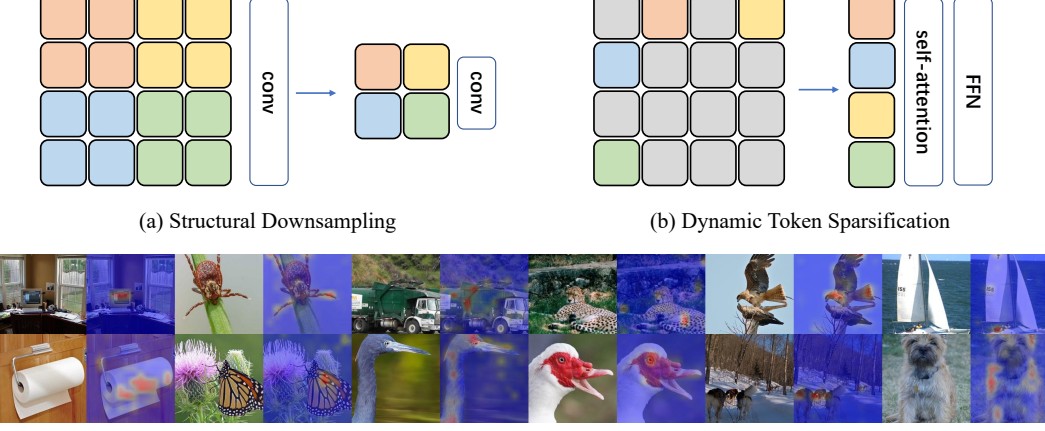

(a) Structural Downsampling

(b) Dynamic Token Sparsification

(c) Attention Visualization

Figure 1: **Illustration of our main idea.** CNN models usually leverage the structural downsampling strategy to build hierarchical architectures as shown in (a). *unstructured* and *data-dependent* downsampling method in (b) can better exploit the sparsity in the input data. Thanks to the nature of the self-attention operation, the unstructured token set is also easy to accelerate through parallel computing. (c) visualizes the impact of each spatial location on the final prediction in the DeiT-S model [25] using the visualization method proposed in [3]. These results demonstrate the final prediction in vision transformers is only based on a subset of most informative tokens, which suggests a large proportion of tokens can be removed without hurting the performance.

downsampling strategy for vision transformers to further leverage the advantages of self-attention (our experiments also show unstructured sparsification can lead to better performance for vision transformers compared to structural downsampling). The basic idea of our method is illustrated in Figure 1.

In this work, we propose to employ a lightweight prediction module to determine which tokens to be pruned in a dynamic way, dubbed as DynamicViT. In particular, for each input instance, the prediction module produces a customized binary decision mask to decide which tokens are uninformative and need to be abandoned. This module is added to multiple layers of the vision transformer, such that the sparsification can be performed in a hierarchical way as we gradually increase the amount of pruned tokens after each prediction module. Once a token is pruned after a certain layer, it will not be ever used in the feed-forward procedure. The additional computational overhead introduced by this lightweight module is quite small, especially considering the computational overhead saved by eliminating the uninformative tokens.

This prediction module can be optimized jointly in an end-to-end manner together with the vision transformer backbone. To this end, two specialized strategies are adopted. The first one is to adopt Gumbel-Softmax [15] to overcome the non-differentiable problem of sampling from a distribution so that it is possible to perform the end-to-end training. The second one is about how to apply this learned binary decision mask to prune the unnecessary tokens. Considering the number of zero elements in the binary decision mask is different for each instance, directly eliminating the uninformative tokens for each input instance during training will make parallel computing impossible. Moreover, this would also hinder the back-propagation for the prediction module, which needs to calculate the probability distribution of whether to keep the token even if it is finally eliminated. Besides, directly setting the abandoned tokens as zero vectors is also not a wise idea since zero vectors will still affect the calculation of the attention matrix. Therefore, we propose a strategy called attention masking where we drop the connection from abandoned tokens to all other tokens in the attention matrix based on the binary decision mask. By doing so, we can overcome the difficulties described above. We also modify the original training objective of the vision transformer by adding a term to constrain the proportion of pruned tokens after a certain layer. During the inference phase, we can directly abandon a fixed amount of tokens after certain layers for each input instance as we no longer need to consider whether the operation is differentiable, and this will greatly accelerate the inference.

We illustrate the effectiveness of our method on ImageNet using DeiT [25] and LV-ViT [16] as backbone. The experimental results demonstrate the competitive trade-off between speed and accuracy. In particular, by hierarchically pruning 66% of the input tokens, we can greatly reduce 31% ~ 37% GFLOPs and improve the throughput by over 40% while the drop of accuracy is within 0.5% for all different vision transformers. Our DynamicViT demonstrates the possibility of exploiting the sparsity in space for the acceleration of transformer-like model. We expect our attempt to open a new path for future work on the acceleration of transformer-like models.

## 2  Related Work

**Vision transformers.**  Transformer model is first widely studied in NLP community [26]. It proves the possibility to use self-attention to replace the recurrent neural networks and their variants. Recent progress has demonstrated the variants of transformers can also be a competitive alternative to CNNs and achieve promising results on different vision tasks including image classification [8, 25, 20, 35, 23], object detection [2], semantic segmentation [34, 5] and 3D analysis [31, 33]. DETR [2] is the first work to apply the transformer model to vision tasks. It formulates the object detection task as a set prediction problem and follows the encoder-decoder design in the transformer to generate a sequence of bounding boxes. ViT [8] is the first work to directly apply transformer architecture on non-overlapping image patches for the image classification task, and the whole framework contains no convolution operation. Compared to CNN-type models, ViT can achieve better performance with large-scale pre-training. It is really preferred if the architecture can achieve the state-of-the-art without any pre-training. DeiT [25] proposes many training techniques so that we can train the convolution-free transformer only on ImageNet1K [7] and achieve better performance than ViT. LV-ViT [16] further improves the performance by introducing a new training objective called token labeling. Both ViT and its follow-ups split the input image into multiple independent image patches and transform these image patches into tokens for further process. This makes it feasible to incorporate the sparsity in space dimension for these transformer-like models.

**Model acceleration.**  Model acceleration techniques are important for the deployment of deep models on edge devices. There are many techniques can be used to accelerate the inference speed of deep model, including quantization [9, 27], pruning [13, 22], low-rank factorization [30], knowledge distillation [14, 19] and so on. There are also many works aims at accelerating the inference speed of transformer models. For example, TinyBERT [17] proposes a distillation method to accelerate the inference of transformer. Star-Transformer [10] reduces quadratic space and time complexity to linear by replacing the fully connected structure with a star-shaped topology. However, all these works focus on NLP tasks, and few works explore the possibility of making use of the characteristic of vision tasks to accelerate vision transformer. Furthermore, the difference between the characteristics of Transformer and CNN also makes it possible to adopt another way for acceleration rather than the methods used for CNN acceleration like filter pruning [13], which removes non-critical or redundant neurons from a deep model. Our method aims at pruning the tokens of less importance instead of the neurons by exploiting the sparsity of informative image patches.

## 3  Dynamic Vision Transformers

### 3.1  Overview

The overall framework of our DynamicViT is illustrated in Figure 2. Our DynamicViT consists of a normal vision transformer as the backbone and several prediction modules. The backbone network can be implemented as a wide range of vision transformer (*e.g.*, ViT [8], DeiT [25], LV-ViT [16]). The prediction modules are responsible for generating the probabilities of dropping/keeping the tokens. The token sparsification is performed hierarchically through the whole network at certain locations. For example, given a 12-layer transformer, we can conduct token sparsification before the 4th, 7th, and 10th blocks. During training, the prediction modules and the backbone network can be optimized in an end-to-end manner thanks to our newly devised attention masking strategy. During inference, we only need to select the most informative tokens according to a predefined pruning ratio and the scores computed by the prediction modules.

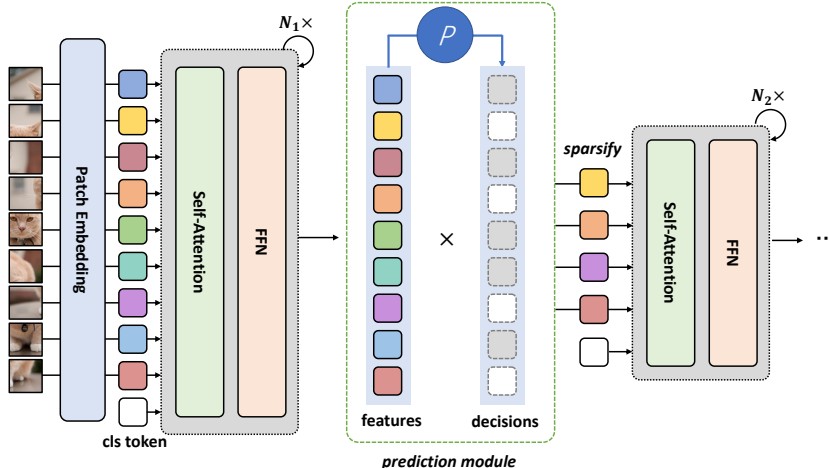

Figure 2: **The overall framework of the proposed approach.** The proposed prediction module is inserted between the transformer blocks to selectively prune less informative token conditioned on features produced by the previous layer. By doing so, less tokens are processed in the followed layers.

## 3.2 Hierarchical Token Sparsification with Prediction Modules

An important characteristic of our DynamicViT is that the token sparsification is performed hierarchically, *i.e.*, we gradually drop the uninformative tokens as the computation proceeds. To achieve this, we maintain a binary decision mask $\hat{\mathbf{D}} \in \{0, 1\}^N$ to indicate whether to drop or keep each token, where $N = HW$ is the number of patch embeddings[2]. We initialize all elements in the decision mask to 1 and update the mask progressively. The prediction modules take the current decision $\hat{\mathbf{D}}$ and the tokens $\mathbf{x} \in \mathbb{R}^{N \times C}$ as input. We first project the tokens using an MLP:

$$\mathbf{z}^{\text{local}} = \text{MLP}(\mathbf{x}) \in \mathbb{R}^{N \times C'}, \tag{1}$$

where $C'$ can be a smaller dimension and we use $C' = C/2$ in our implementation. Similarly, we can compute a global feature by:

$$\mathbf{z}^{\text{global}} = \text{Agg}(\text{MLP}(\mathbf{x}), \hat{\mathbf{D}}) \in \mathbb{R}^{C'}, \tag{2}$$

where $\text{Agg}$ is the function which aggregate the information all the existing tokens and can be simply implemented as an average pooling:

$$\text{Agg}(\mathbf{u}, \hat{\mathbf{D}}) = \frac{\sum_{i=1}^{N} \hat{\mathbf{D}}_i \mathbf{u}_i}{\sum_{i=1}^{N} \hat{\mathbf{D}}_i}, \quad \mathbf{u} \in \mathbb{R}^{N \times C'}. \tag{3}$$

The local feature encodes the information of a certain token while the global feature contains the context of the whole image, thus both of them are informative. Therefore, we combine both the local and global features to obtain local-global embeddings and feed them to another MLP to predict the probabilities to drop/keep the tokens:

$$\mathbf{z}_i = [\mathbf{z}_i^{\text{local}}, \mathbf{z}_i^{\text{global}}], \quad 1 \leq i \leq N, \tag{4}$$

$$\boldsymbol{\pi} = \text{Softmax}(\text{MLP}(\mathbf{z})) \in \mathbb{R}^{N \times 2}, \tag{5}$$

where $\boldsymbol{\pi}_{i,0}$ denotes the probability of dropping the $i$-th token and $\boldsymbol{\pi}_{i,1}$ is the probability of keeping it. We can then generate current decision $\mathbf{D}$ by sampling from $\boldsymbol{\pi}$ and update $\hat{\mathbf{D}}$ by

$$\hat{\mathbf{D}} \leftarrow \hat{\mathbf{D}} \odot \mathbf{D}, \tag{6}$$

where $\odot$ is the Hadamard product, indicating that once a token is dropped, it will never be used.

---

[2]We omit the class token for simplicity, while in practice we always keep the class token (*i.e.*, the decision for class token is always "1").

## 3.3 End-to-end Optimization with Attention Masking

Although our target is to perform token sparsification, we find it non-trivial to implement in practice during training. First, the sampling from $\boldsymbol{\pi}$ to get binary decision mask $\mathbf{D}$ is is non-differentiable, which impedes the end-to-end training. To overcome this, we apply the Gumbel-Softmax technique [15] to sample from the probabilities $\boldsymbol{\pi}$:

$$\mathbf{D} = \text{Gumbel-Softmax}(\boldsymbol{\pi})_{*,1} \in \{0,1\}^N, \tag{7}$$

where we use the index "1" because $\mathbf{D}$ represents the mask of the *kept* tokens. The output of Gumbel-Softmax is a one-hot tensor, of which the expectation equals $\boldsymbol{\pi}$ exactly. Meanwhile, Gumbel-Softmax is differentiable thus makes it possible for end-to-end training.

The second obstacle comes when we try to prune the tokens during training. The decision mask $\hat{\mathbf{D}}$ is usually unstructured and the masks for different samples contain various numbers of 1's. Therefore, simply discarding the tokens where $\hat{\mathbf{D}}_i = 0$ would result in a non-uniform number of tokens for samples within a batch, which makes it hard to parallelize the computation. Thus, we must keep the number of tokens unchanged, while cut down the interactions between the pruned tokens and other tokens. We also find that merely zero-out the tokens to be dropped using the binary mask $\hat{\mathbf{D}}$ is not feasible, because in the calculation of self-attention matrix [26]

$$\mathbf{A} = \text{Softmax}\left(\frac{\mathbf{Q}\mathbf{K}^T}{\sqrt{C}}\right) \tag{8}$$

the zeroed tokens will still influence other tokens through the $\text{Softmax}$ operation. To this end, we devise a strategy called attention masking which can totally eliminate the effects of the dropped tokens. Specifically, we compute the attention matrix by:

$$\mathbf{P} = \mathbf{Q}\mathbf{K}^T/\sqrt{C} \in \mathbb{R}^{N \times N}, \tag{9}$$

$$\mathbf{G}_{ij} = \begin{cases} 1, & i = j, \\ \hat{\mathbf{D}}_j, & i \neq j. \end{cases} \qquad 1 \leq i, j \leq N, \tag{10}$$

$$\tilde{\mathbf{A}}_{ij} = \frac{\exp(\mathbf{P}_{ij})\mathbf{G}_{ij}}{\sum_{k=1}^N \exp(\mathbf{P}_{ik})\mathbf{G}_{ik}}, \qquad 1 \leq i, j \leq N. \tag{11}$$

By Equation (10) we construct a graph where $\mathbf{G}_{ij} = 1$ means the $j$-th token will contribute to the update of the $i$-th token. Note that we explicitly add a self-loop to each token to improve numerically stability. It is also easy to show the self-loop does not influence the results: if $\hat{\mathbf{D}}_j = 0$, the $j$-th token will not contribute to any tokens other than itself. Equation (11) computes the masked attention matrix $\tilde{\mathbf{A}}$, which is equivalent to the attention matrix calculated by considering only the kept tokens but has a constant shape $N \times N$ during training.

## 3.4 Training and Inference

We now describe the training objectives of our DynamicViT. The training of DynamicViT includes training the prediction modules such that they can produce favorable decisions and fine-tuning the backbone to make it adapt to token sparsification. Assuming we are dealing with a minibatch of $B$ samples, we adopt the standard cross-entropy loss:

$$\mathcal{L}_{\text{cls}} = \text{CrossEntropy}(\mathbf{y}, \bar{\mathbf{y}}), \tag{12}$$

where $\mathbf{y}$ is the prediction of the DynamicViT (after softmax) and $\bar{\mathbf{y}}$ is the ground truth.

To minimize the influence on performance caused by our token sparsification, we use the original backbone network as a teacher model and hope the behavior of our DynamicViT as close to the teacher model as possible. Specifically, we consider this constraint from two aspects. First, we make the finally remaining tokens of the DynamicViT close to the ones of the teacher model, which can be viewed as a kind of self-distillation:

$$\mathcal{L}_{\text{distill}} = \frac{1}{\sum_{b=1}^B \sum_{i=1}^N \hat{\mathbf{D}}_i^{b,S}} \sum_{b=1}^B \sum_{i=1}^N \hat{\mathbf{D}}_i^{b,S}(\mathbf{t}_i - \mathbf{t}_i')^2, \tag{13}$$

Table 1: **Main results on ImageNet.** We apply our method on three representative vision transformers: DeiT-S, LV-ViT-S and LV-ViT-M. DeiT-S [25] is a widely used vision transformer with the simple architecture. LV-ViT-S and LV-ViT-M [16] are the state-of-the-art vision transformers. We report the top-1 classification accuracy, theoretical complexity in FLOPs and throughput for different ratio $\rho$. The throughput is measured on a single NVIDIA RTX 3090 GPU with batch size fixed to 32.

| Base Model | Metrics | Keeping Ratio $\rho$ at each stage | | | |
| --- | --- | --- | --- | --- | --- |
| | | 1.0 | 0.9 | 0.8 | 0.7 |
| DeiT-S [25] | ImageNet Acc. (%) | 79.8 | 79.8 (-0.0) | 79.6 (-0.2) | 79.3 (-0.5) |
| | GFLOPs | 4.6 | 4.0 (-14%) | 3.4 (-27%) | 2.9 (-37%) |
| | Throughput (im/s) | 1337.7 | 1524.8 (+14%) | 1774.6 (+33%) | 2062.1 (+54%) |
| LV-ViT-S [16] | ImageNet Acc. (%) | 83.3 | 83.3 (-0.0) | 83.2 (-0.1) | 83.0 (-0.3) |
| | GFLOPs | 6.6 | 5.8 (-12%) | 5.1 (-22%) | 4.6 (-31%) |
| | Throughput (im/s) | 993.3 | 1108.3 (+12%) | 1255.6 (+26%) | 1417.6 (+43%) |
| LV-ViT-M [16] | ImageNet Acc. (%) | 84.0 | 83.9 (-0.1) | 83.9 (-0.1) | 83.8 (-0.2) |
| | GFLOPs | 12.7 | 11.1 (-13%) | 9.6 (-24%) | 8.5 (-33%) |
| | Throughput (im/s) | 589.5 | 688.5 (+17%) | 791.2 (+34%) | 888.2 (+50%) |

where $\mathbf{t}_i$ and $\mathbf{t}_i'$ denotes the $i$-th token after the last block of the DynamicViT and the teacher model, respectively. $\hat{\mathbf{D}}^{b,s}$ is the decision mask for the $b$-th sample at the $s$-th sparsification stage. Second, we minimize the difference of the predictions between our DynamicViT and its teacher via the KL divergence:

$$\mathcal{L}_{\mathrm{KL}} = \mathrm{KL}\left(\mathbf{y}\|\mathbf{y}'\right), \tag{14}$$

where $\mathbf{y}'$ is the prediction of the teacher model.

Finally, we want to constrain the ratio of the kept tokens to a predefined value. Given a set of target ratios for $S$ stages $\boldsymbol{\rho} = [\rho^{(1)}, \ldots, \rho^{(S)}]$, we utilize an MSE loss to supervise the prediction module:

$$\mathcal{L}_{\mathrm{ratio}} = \frac{1}{BS}\sum_{b=1}^{B}\sum_{s=1}^{S}\left(\rho^{(s)} - \frac{1}{N}\sum_{i=1}^{N}\hat{\mathbf{D}}_i^{b,s}\right)^2. \tag{15}$$

The full training objective is a combination of the above objectives:

$$\mathcal{L} = \mathcal{L}_{\mathrm{cls}} + \lambda_{\mathrm{KL}}\mathcal{L}_{\mathrm{KL}} + \lambda_{\mathrm{distill}}\mathcal{L}_{\mathrm{distill}} + \lambda_{\mathrm{ratio}}\mathcal{L}_{\mathrm{ratio}}, \tag{16}$$

where we set $\lambda_{\mathrm{KL}} = 0.5, \lambda_{\mathrm{distill}} = 0.5, \lambda_{\mathrm{ratio}} = 2$ in all our experiments.

During inference, given the target ratio $\boldsymbol{\rho}$, we can directly discard the less informative tokens via the probabilities produced by the prediction modules such that only exact $m^s = \lfloor \rho^s N \rfloor$ tokens are kept at the $s$-th stage. Formally, for the $s$-th stage, let

$$\mathcal{I}^s = \mathrm{argsort}(\boldsymbol{\pi}_{*,1}) \tag{17}$$

be the indices sorted by the keeping probabilities $\boldsymbol{\pi}_{*,1}$, we can then keep the tokens of which the indices lie in $\mathcal{I}^s_{1:m^s}$ while discarding the others. In this way, our DynamicViT prunes less informative tokens dynamically at runtime, thus can reduce the computational costs during inference.

## 4 Experimental Results

In this section, we will demonstrate the superiority of the proposed DynamicViT through extensive experiments. In all of our experiments, we fix the number of sparsification stages $S = 3$ and apply the target keeping ratio $\boldsymbol{\rho}$ as a geometric sequence $[\rho, \rho^2, \rho^3]$ where $\rho$ ranges from $(0, 1)$. During training DynamicViT models, we follow most of the training techniques used in DeiT [25]. We use the pre-trained vision transformer models to initialize the backbone models and jointly train the whole model for 30 epochs. We set the learning rate of the prediction module to $\frac{\mathrm{batch\ size}}{1024} \times 0.001$ and use $0.01\times$ learning rate for the backbone model. We fix the weights of the backbone models in the first 5 epochs. All of our models are trained on a single machine with 8 GPUs. Other training setups and details can be found in the supplementary material.

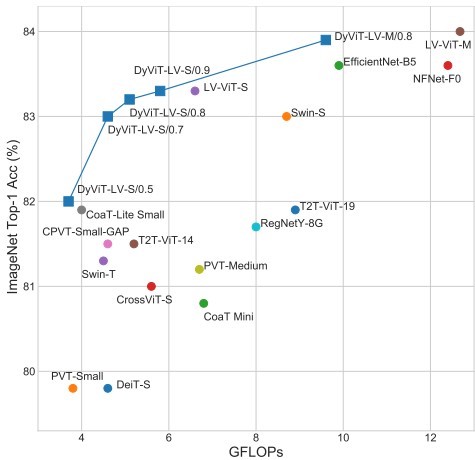
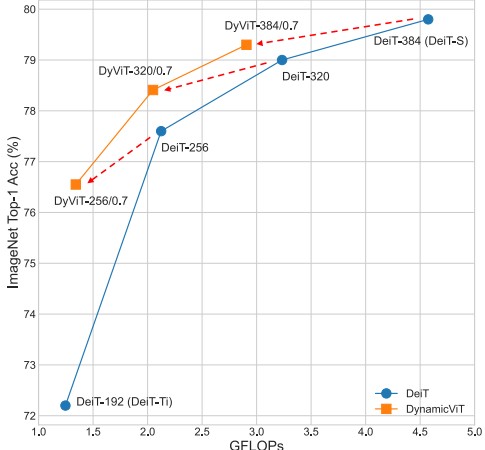

Figure 3: Model complexity (FLOPs) and top-1 accuracy trade-offs on ImageNet. We compare DynamicViT with the state-of-the-art image classification models. Our models achieve better trade-offs compared to the various vision transformers as well as carefully designed CNN models.

Figure 4: Comparison of our dynamic token sparsification method with model width scaling. We train our DynamicViT based on DeiT models with embedding dimension varying from 192 to 384 and fix ratio $\rho = 0.7$. We see dynamic token sparsification is more efficient than commonly used model width scaling.

## 4.1 Main results

One of the most advantages of the DynamicViT is that it can be applied to a wide range of vision transformer architectures to reduce the computational complexity with minor loss of performance. In Table 1, we summarize the main results on ImageNet [7] where we evaluate our DynamicViT used three base models (DeiT-S [25], LV-ViT-S [16] and LV-ViT-M [16]). We report the top-1 accuracy, FLOPs, and the throughput under different keeping ratios $\rho$. Note that our token sparsification is performed hierarchically in three stages, there are only $\lfloor N\rho^3 \rfloor$ tokens left after the last stage. The throughput is measured on a single NVIDIA RTX 3090 GPU with batch size fixed to 32. We demonstrate that our DynamicViT can reduce the computational costs by $31\% \sim 37\%$ and accelerate the inference at runtime by $43\% \sim 54\%$, with the neglectable influence of performance $(-0.2\% \sim -0.5\%)$.

## 4.2 Comparisons with the-state-of-the-arts

In Table 2, we compare the DynamicViT with the state-of-the-art models in image classification, including convolutional networks and transformer-like architectures. We use the DynamicViT with LV-ViT [16] as the base model and use the "$/\rho$" to indicate the keeping ratio. We observe that our DynamicViT exhibits favorable complexity/accuracy trade-offs at all three complexity levels. Notably, we find our DynamicViT-LV-M/0.7 beats the EfficientNet-B5 [24] and NFNet-F0 [1], which are two of the current state-of-the-arts CNN architectures. This can also be shown clearer in Figure 3, where we plot the FLOPS-accuracy curve of DynamicViT series (where we use DyViT for short), along with other state-of-the-art models. We can also observe that DynamicViT can achieve better trade-offs than LV-ViT series, which strongly demonstrates the effectiveness of our method.

## 4.3 Analysis

**DynamicViT for model scaling.** The success of EfficientNet [24] shows that we can obtain a model with better complexity/accuracy tradeoffs by scaling the model along different dimensions. While in vision transformers, the most commonly used method to scale the model is to change the number of channels, our DynamicViT provides another powerful tool to perform token sparsification. We analysis this nice property of DynamicViT in Figure 4. First, we train several DeiT [25] models with the embedding dimension varying from 192 (DeiT-Ti) to 384 (DeiT-S). Second, we train our DynamicViT based on those models with the keeping ratio $\rho = 0.7$. We find that after performing token sparsification, the complexity of the model is reduced to be similar to its variant with a smaller embedding dimension. Specifically, we observe that by applying our DynamicViT to DeiT-256, we

Table 2: **Comparisons with the state-of-the-arts on ImageNet.** We compare our DynamicViT models with state-of-the-art image classifciation models with comparable FLOPs and number of parameters. We use the DynamicViT with LV-ViT [16] as the base model and use the "/$\rho$" to indicate the keeping ratio. We also include the results of LV-ViT models as references.

| Model | Params (M) | GFLOPs | Resolution | Top-1 Acc (%) |
|---|---|---|---|---|
| DeiT-S [25] | 22.1 | 4.6 | 224 | 79.8 |
| PVT-Small [28] | 24.5 | 3.8 | 224 | 79.8 |
| CoaT Mini [29] | 10.0 | 6.8 | 224 | 80.8 |
| CrossViT-S [4] | 26.7 | 5.6 | 224 | 81.0 |
| PVT-Medium [28] | 44.2 | 6.7 | 224 | 81.2 |
| Swin-T [20] | 29.0 | 4.5 | 766 | 81.3 |
| T2T-ViT-14 [32] | 22.0 | 5.2 | 224 | 81.5 |
| CPVT-Small-GAP [6] | 23.0 | 4.6 | 817 | 81.5 |
| CoaT-Lite Small [29] | 20.0 | 4.0 | 224 | 81.9 |
| LV-ViT-S [16] | 26.2 | 6.6 | 224 | 83.3 |
| DynamicViT-LV-S/0.5 | 26.9 | 3.7 | 224 | 82.0 |
| DynamicViT-LV-S/0.7 | 26.9 | 4.6 | 224 | 83.0 |
| RegNetY-8G [21] | 39.0 | 8.0 | 224 | 81.7 |
| T2T-ViT-19 [32] | 39.2 | 8.9 | 224 | 81.9 |
| Swin-S [20] | 50.0 | 8.7 | 224 | 83.0 |
| EfficientNet-B5 [24] | 30.0 | 9.9 | 456 | 83.6 |
| NFNet-F0 [1] | 72.0 | 12.4 | 256 | 83.6 |
| DynamicViT-LV-M/0.7 | 57.1 | 8.5 | 224 | 83.8 |
| ViT-Base/16 [8] | 86.6 | 17.6 | 224 | 77.9 |
| DeiT-Base/16 [25] | 86.6 | 17.6 | 224 | 81.8 |
| CrossViT-B [4] | 104.7 | 21.2 | 224 | 82.2 |
| T2T-ViT-24 [32] | 64.1 | 14.1 | 224 | 82.3 |
| TNT-B [11] | 66.0 | 14.1 | 224 | 82.8 |
| RegNetY-16G [21] | 84.0 | 16.0 | 224 | 82.9 |
| Swin-B [20] | 88.0 | 15.4 | 224 | 83.3 |
| LV-ViT-M [16] | 55.8 | 12.7 | 224 | 84.0 |
| DynamicViT-LV-M/0.8 | 57.1 | 9.6 | 224 | 83.9 |

obtain a model that has a comparable computational complexity to DeiT-Ti, but enjoys around $4.3\%$ higher ImageNet top-1 accuracy.

**Visualizations.** To further investigate the behavior of DynamicViT, we visualize the sparsification procedure in Figure 5. We show the original input image and the sparsification results after the three stages, where the masks represent the corresponding tokens are discarded. We find that through the hierarchically token sparsification, our DynamicViT can gradually drop the uninformative tokens and finally focus on the objects in the images. This phenomenon also suggests that the DynamicViT leads to better interpretability, *i.e.*, it can locate the important parts in the image which contribute most to the classification step-by-step.

Besides the sample-wise visualization we have shown above, we are also interested in the statistical characteristics of the sparsification decisions, *i.e.*, what kind of general patterns does the DynamicViT learn from the dataset? We then use the DynamicViT to generate the decisions for all the images in the ImageNet validation set and compute the keep probability of each token in all three stages, as shown in Figure 6. We average pool the probability maps into $7 \times 7$ such that they can be visualized more easily. Unsurprisingly, we find the tokens in the middle of the image tend to be kept, which is reasonable because in most images the objects are located in the center. We can also find that the later stage generally has lower probabilities to be kept, mainly because that the keeping ratio at the $s$ stage is $\rho^s$, which decreases exponentially as $s$ increases.

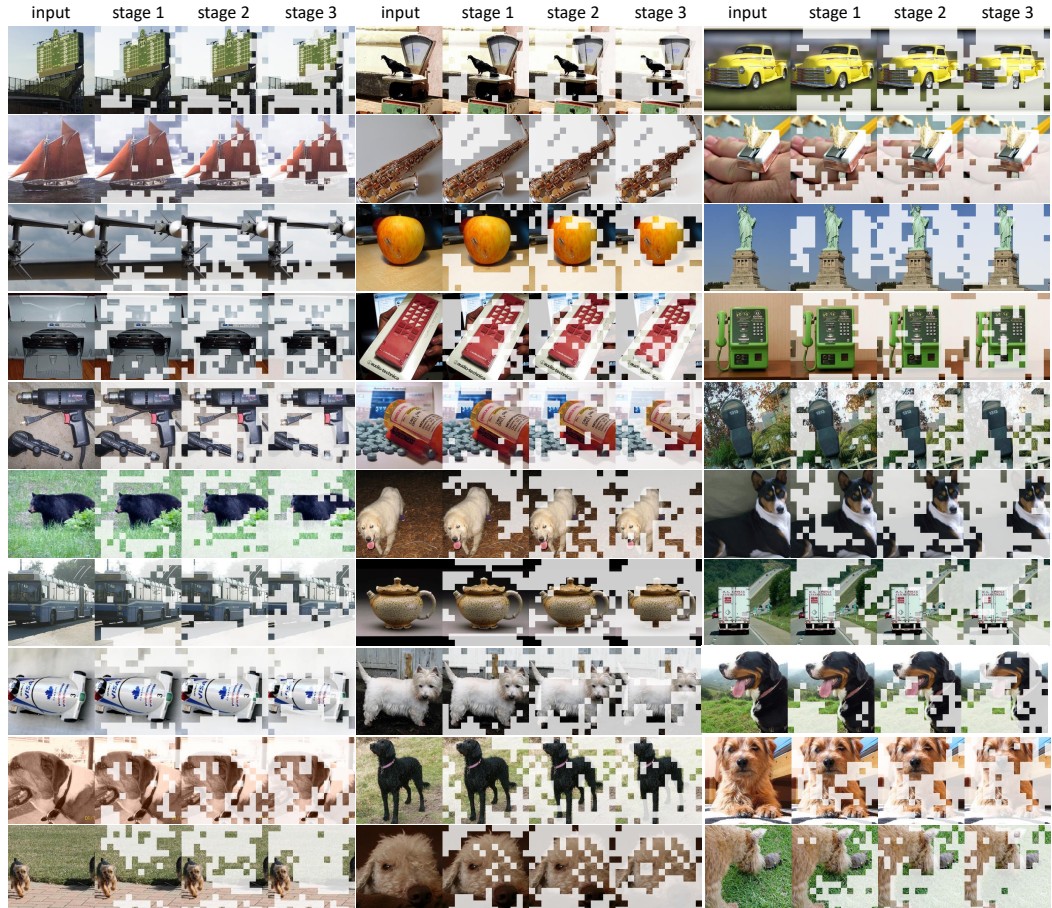

| input | stage 1 | stage 2 | stage 3 | input | stage 1 | stage 2 | stage 3 | input | stage 1 | stage 2 | stage 3 |

Figure 5: **Visualization of the progressively sparsified tokens.** We show the original input image and the sparsification results after the three stages, where the masks represent the corresponding tokens are discarded. We see our method can gradually focus on the most representative regions in the image. This phenomenon suggests that the DynamicViT has better interpretability.

Table 3: Effects of different losses. We provide the results after removing the distillation loss and the KL loss.

| Base Model | DeiT-S | LVViT-S |
|---|---|---|
| DynamicViT | 79.3(-0.5) | 83.0(-0.3) |
| w/o distill (Eq.13) | 79.3(-0.5) | 82.7(-0.6) |
| w/o KL (Eq.14) | 79.2(-0.6) | 82.9(-0.4) |
| w/o distill & KL | 79.2(-0.6) | 82.5(-0.8) |

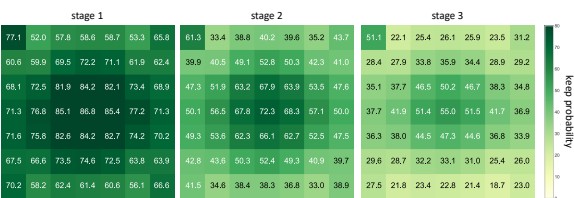

Figure 6: The keep probabilities of the tokens at each stage.

**Effects of different losses.** We show the effects of different losses in Table 3. We see the improvement brought by the distillation loss and the KL loss is not very significant, but it can consistently further boost the performance of various models.

**Comparisons of different sparsification strategies.** As illustrated in Figure 2, the dynamic token sparsification is unstructured. To discuss whether the dynamic sparsification is better than other strategies, we perform ablation experiments and the results are shown in Table 4. For the structural downsampling, we perform an average pooling with kernel size $2 \times 2$ after the sixth block of the baseline DeiT-S [25] model, which has similar FLOPs to our DynamicViT. The static token sparsification means that the sparsification decisions are not conditioned on the input tokens. We also compare our method with other token removal methods like randomly removing tokens or removing

Table 4: Comparisons of different sparsification strategies. We investigate different methods to select redundant tokens based on the DeiT-S model. We report the top-1 accuracy on ImageNet for different methods. We fix the complexity of the accelerated models to 2.9G FLOPs for fair comparisons.

(a) Dynamic sparsification *vs.* static/structural downsampling.

| Model | Acc. (%) |
|---|---|
| Structural | 78.2 (-1.6) |
| Static | 73.4 (-6.4) |
| Dynamic | 79.3 (-0.5) |

(b) Different redundant token removal methods.

| Model | Acc. (%) |
|---|---|
| Random | 77.5 (-2.3) |
| Attention | 78.1 (-1.7) |
| Prediction | 79.3 (-0.5) |

(c) Effects of number of sparsification stages.

| Model | Acc. (%) |
|---|---|
| Single-stage | 77.4 (-2.4) |
| Two-stage | 79.2 (-0.6) |
| Three-stage | 79.3 (-0.5) |

Table 5: Results on larger models. We apply our method to the model with larger width (*i.e.*, DeiT-B) and the model with larger input size (*i.e.*, DeiT-S with $384 \times 384$ input).

(a) Results on DeiT-B.

| Model | GFLOPs | Acc. (%) |
|---|---|---|
| DeiT-B | 17.5 | 81.8 |
| DynamicViT-B/0.7 | 11.2 (-36%) | 81.3 (-0.5) |

(b) Results on the $384 \times 384$ input.

| Model | GFLOPs | Acc. (%) |
|---|---|---|
| DeiT-S | 15.5 | 81.6 |
| DynamicViT-S/0.7 | 9.5 (-39%) | 81.4 (-0.2) |
| DynamicViT-S/0.5 | 7.0 (-55%) | 80.3 (-1.3) |

tokens based the attention score of the class token. We find through the experiments that although other strategies have similar computational complexities, the proposed dynamic token sparsification method achieves the best accuracy. We also show that the progressive sparsification method is significantly better than one-stage sparsification.

**Accelerating larger models.** To show the effectiveness of our method on larger models, we apply our method to the model with larger width (*i.e.*, DeiT-B) and models with larger input size (*i.e.*, DeiT-S with $384 \times 384$ input). The results are presented in Table 5. We see our method also works well on the larger DeiT model. The accuracy drop become less significant when we apply our method to the model with larger feature maps. Notably, we can reduce the complexity of the DeiT-S model with $384 \times 384$ input by over 50% with only 1.3% accuracy drop.

## 5 Conclusion

In this work, we open a new path to accelerate vision transformer by exploiting the sparsity of informative patches in the input image. For each input instance, our DynamicViT model prunes the tokens of less importance in a dynamic way according to the customized binary decision mask output from the lightweight prediction module, which fuses the local and global information containing in the tokens. The prediction module is added to multiple layers such that the token pruning is performed in a hierarchical way. Gumbel-Softmax and attention masking techniques are also incorporated for the end-to-end training of the transformer model together with the prediction module. During the inference phase, our approach can greatly improves the efficiency by gradually pruning 66% of the input tokens, while the drop of accuracy is less than 0.5% for different transformer backbone. In this paper, we focus on the image classification task. Extending our method to other scenarios like video classification and dense prediction tasks can be interesting directions.

**Acknowledgment**

This work was supported in part by the National Key Research and Development Program of China under Grant 2017YFA0700802, in part by the National Natural Science Foundation of China under Grant 62125603, Grant 61822603, Grant U1813218, Grant U1713214, in part by Beijing Academy of Artificial Intelligence (BAAI), in part by National Science Foundation under grant IIS-1901527, IIS-2008173, IIS-2048280, and in part by a grant from the Institute for Guo Qiang, Tsinghua University.

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
