# DynamicViT: Efficient Vision Transformers with Dynamic Token Sparsification
## *Supplementary Material*

## A  Implementation Details

We conduct our experiments on the ImageNet (also known as ILSVRC2012) [1] dataset. ImageNet is a commonly used benchmark for image classification. We train our models on the training set, which consists of 1.28M images. The top-1 accuracy is measured on the 50k validation images following common practice [2, 4]. To fairly compare with previous methods, we report the single crop results.

We fix the number of sparsification stages $S = 3$ in all of our experiments, since this setting can lead to a decent trade-off between complexity and performance. For the sake of simplicity, we set the target keeping ratio $\boldsymbol{\rho}$ as a geometric sequence $[\rho, \rho^2, \rho^3]$, where $\rho$ is the keeping ratio after each sparsifcation ranging from $(0, 1)$. For the prediction module, we use the identical architecture for different stages. We use two $\texttt{LayerNorm} \rightarrow \texttt{Linear}(C, C/2) \rightarrow \texttt{GELU}$ block to produce $\mathbf{z}^{\text{local}}$ and $\mathbf{z}^{\text{global}}$ respectively. We employ a $\texttt{Linear}(C, C/2) \rightarrow \texttt{GELU} \rightarrow \texttt{Linear}(C/2, C/4) \rightarrow \texttt{GELU} \rightarrow \texttt{Linear}(C/4, 2) \rightarrow \texttt{Softmax}$ block to predict the probabilities.

During training our DynamicViT models, we follow most of the training techniques used in DeiT [4]. We use the pre-trained vision transformer models to initialize the backbone models and jointly train the backbone model as well as the prediction modules for 30 epochs. We set the learning rate of the prediction module to $\frac{\text{batch size}}{1024} \times 0.001$ and use $0.01\times$ learning rate for the backbone model. The batch size is adjusted adaptively for different models according to the GPU memory. We fix the weights of the backbone models in the first 5 epochs. All of our models can be trained on a single machine with 8 NVIDIA GTX 1080Ti GPUs.

## B  More Analysis

In this section, we provide more analysis of our method. We investigate the effects of progressive sparsification, distillation loss, ratio loss, and keeping ratio. We also include more visualization results. The following describes the details of the experiments, results and analysis.

**Progressive sparsification.**    To verify the effectiveness of the progressive sparsification strategy, we test different sparsification methods that result in similar overall complexity. Here we provide more detailed results and more analysis. We find that progressive sparsification is much better than single-shot sparsification. Increasing the number of stages will lead to better performance. Since further increasing the number of stages ($> 3$) will not lead to significantly better performance but add computation, we use a 3-stage progressive sparsification strategy in our main experiments.

**Ablation on the distillation loss and ratio loss.**    The weights of the distillation losses and ratio loss are the key hyper-parameters in our method. Since the token-wise distillation loss and the KL divergence loss play similar roles in our method, we set $\lambda_{\text{KL}} = \lambda_{\text{distill}}$ in all of our experiments for the sake of simplicity. In this experiment, we fix the keeping ratio $\rho$ to be 0.7. We find our method is not sensitive to these hyper-parameters in general. The proposed ratio loss can encourage the

35th Conference on Neural Information Processing Systems (NeurIPS 2021).

|  | Top-1 accuracy (%) | GFLOPs |
|---|---|---|
| DeiT-S [4] | 79.8 | 4.6 |
| $\rho = 0.25$, $[\rho]$ (single-stage) | 77.4(-2.4) | 2.9(-37%) |
| $\rho = 0.60$, $[\rho, \rho^2]$ (two-stage) | 79.2(-0.6) | 2.9(-37%) |
| $\rho = 0.70$, $[\rho, \rho^2, \rho^3]$ (three-stage) | 79.3(-0.5) | 2.9(-37%) |

model to reach the desired acceleration rate. Distillation losses can improve the performance after sparsification. We directly apply the best hyper-parameters searched on DeiT-S for all models.

|  | Top-1 accuracy (%) |
|---|---|
| DeiT-S [4] | 79.8 |
| $\lambda_{\mathrm{KL}} = \lambda_{\mathrm{distill}} = 0$ | 79.17(-0.63) |
| $\lambda_{\mathrm{KL}} = \lambda_{\mathrm{distill}} = 0.5$ | 79.32(-0.48) |
| $\lambda_{\mathrm{KL}} = \lambda_{\mathrm{distill}} = 1$ | 79.23(-0.57) |

|  | Top-1 accuracy (%) |
|---|---|
| DeiT-S [4] | 79.8 |
| $\lambda_{\mathrm{ratio}} = 1$ | 79.15(-0.65) |
| $\lambda_{\mathrm{ratio}} = 2$ | 79.32(-0.48) |
| $\lambda_{\mathrm{ratio}} = 4$ | 79.29(-0.51) |

**Smaller keeping ratio.** We have also tried applying a smaller keeping ratio (larger acceleration rate). The results based on DeiT-S [4] and LV-ViT-S [3] models are presented in the following tables. We see that using $\rho < 0.7$ will lead to a significant accuracy drop while reducing fewer FLOPs. Since only 22% and 13% tokens are remaining in the last stage when we set $\rho$ to 0.6 and 0.5 respectively, small $\rho$ may cause a significant information loss. Therefore, we use $\rho \geq 0.7$ in our main experiments. Jointly scaling $\rho$ and the model width can be a better solution to achieve a large acceleration rate as shown in Figure 4 in the paper.

|  | Top-1 acc. (%) | GFLOPs |
|---|---|---|
| DeiT-S [4] | 79.8 | 4.6 |
| $\rho = 0.9$ | 79.8(-0.0) | 4.0(-14%) |
| $\rho = 0.8$ | 79.6(-0.3) | 3.4(-27%) |
| $\rho = 0.7$ | 79.3(-0.5) | 2.9(-37%) |
| $\rho = 0.6$ | 78.5(-1.3) | 2.5(-46%) |
| $\rho = 0.5$ | 77.5(-2.3) | 2.2(-52%) |

|  | Top-1 acc. (%) | GFLOPs |
|---|---|---|
| LV-ViT-S [3] | 83.3 | 6.6 |
| $\rho = 0.9$ | 83.3(-0.0) | 5.8(-12%) |
| $\rho = 0.8$ | 83.2(-0.1) | 5.1(-22%) |
| $\rho = 0.7$ | 83.0(-0.3) | 4.6(-31%) |
| $\rho = 0.6$ | 82.6(-0.7) | 4.1(-38%) |
| $\rho = 0.5$ | 82.0(-1.3) | 3.7(-44%) |

**More visual results.** We provide more visual results in Figure 1. The input images are randomly sampled from the validation set of ImageNet. We see our method works well for different images from various categories.

input stage1 stage2 stage3 input stage1 stage2 stage3 input stage1 stage2 stage3

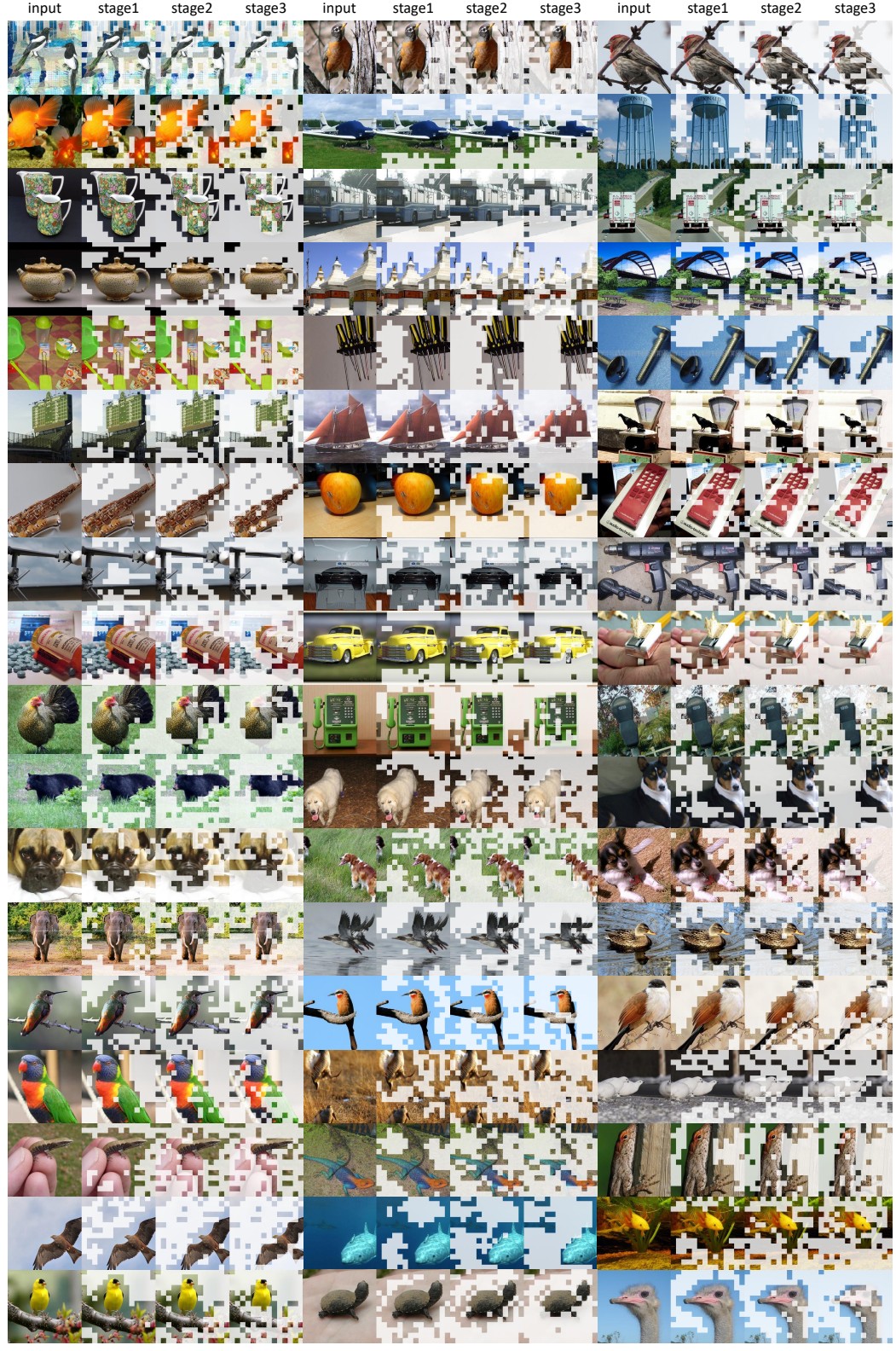

Figure 1: **More visual results**. The input images are randomly sampled from the validation set of ImageNet. We see our method works well for different images from various categories.