# OpenReview forum: "DynamicViT: Efficient Vision Transformers with Dynamic Token Sparsification"
_NeurIPS.cc/2021/Conference — NeurIPS 2021 Poster_

### Official Review · Reviewer_m39c · 2021-07-12

**Rating:** 6
**Confidence:** 4

**Summary:**

A token sparsification method is proposed in this paper for speeding up ViT. Attention masking is added into the ViT framework with learnable dropping out via back-propagations.  Unessential tokens are gradually pruned via attention masking for inference acceleration. The modified ViT, named dynamicViT, maintains the major accuracy while reducing one third FLOPs.

**Limitations And Societal Impact:**

See above. The negative societal impact is not discussed. The checklist is shown in the supplementary file, which does not follow the guideline that it must appear in the submitted PDF, immediately after references.

**Main Review:**

The main idea of this work is to develop an attention mask strategy for discarding unimportant tokens. The importance is validated by the attention mask which is learnable via gradient back-propagations. This design functions as a module that is integrated into ViTs. It splits one ViT into two parts. The first part maintains the original operations of ViT while the second part proceeds with reduced tokens.  During training, a knowledge distillation based method is introduced to train dynamicViT based on the original ViT.

The whole framework is to reduce the tokens while maintaining accuracy. This seems overall fine. There are a few concerns raised regarding the methodology, presentation, and experiments.

1. In Fig. 2, there are $N_1$ blocks before sparsification and there are $N_2$ blocks afterward. It is not clear how $N_1$ and $N_2$ are set according to different transformers.  Moreover, it seems a tradeoff to set $N_1$ and $N_2$ for acceleration and accuracy.

2. The proposed framework is posed as a knowledge distillation problem, which is specified in Sec. 3.4. It raises an interesting thinking that this approach aims at reducing tokens rather than reducing network parameters (e.g., linear projection parameters). It sounds like DeiT and DynamicViT function similarly (i.e., pruning network params v.s. pruning tokens). A discussion upon DeiT would be helpful to improve the motivation.

3. In the experiments, dynamicViT only focuses on LV-VIT architecture. As the proposed method seems general, is it suitable for different transformers? If so, how? (see q1).

4. The attention mask is learned by gradient back-propagation. How the computed loss will affect this mask to discard unimportant tokens. More explanation on this aspect would make the technique more convincing. For instance, how the mask looks like? What if the majority value of the mask is 0? Since the mask is influenced by the gradients, the values of the mask are sometimes not satisfactory. Have the authors considered any regularization upon that?

5. Eq. 9-11 seems to overcomplicate the zero token issue. Reshaping the remaining tokens to make them compact is a way to ensure the following computations.

----------------------------------------------post rebuttal---------------------------------------------

While the proposed token sparsification seems not suitable for SWIN-based transformers where the tokens are densely organized for further inference. The authors have addressed most of the concerns. This reviewer turns positive on the current submission.

**Time Spent Reviewing:**

4

---

> ### Author Response · Authors · 2021-08-10
> **Response to Reviewer m39c**
>
> We would like to thank the reviewer for the careful reading and valuable comments. We address the questions and clarify the issues accordingly as described below.
>
>
> **Q1: The setting of $N_1,N_2$**
>
> **[Reply]** Sorry for the confusion. For the sake of simplicity, we use the same number of blocks between the sparsification stages in all the experiments. For example, DeiT has 12 blocks in total so we perform the token sparsification 3 times uniformly and set $N_1=N_2=N_3=N_4=3$. We will elaborate on this in the revised paper.
>
> **Q2: About DeiT**
>
> **[Reply]** We agree that DeiT is an important and widely used architecture in vision transformers. However, as far as we know, DeiT applies a better training strategy (regularization and distillation) to improve the performance of ViT, with both the parameters and computational costs **unchanged**. On the other hand, our method aims to prune less informative tokens to reduce the computation. Therefore, our contribution is orthogonal to that of DeiT. We have also conducted experiments using DeiT as the baseline (Table 1) and demonstrated our method can improve the throughput significantly with a minor accuracy drop.
>
> **Q3: Application of DynamicViT to different transformers**
>
> **[Reply]** The proposed token sparsification is a method that is designed for self-attention. Therefore, it can be applied to any transformer architecture that is built up with self-attention layers. We can simply add some sparsification stages through the whole computation and prune the tokens before the self-attention layer.
>
> **Q4: About the attention mask**
>
> **[Reply]** The attention mask is generated by the Gumbel-Softmax technique, which can produce binary output while enabling the gradient back-propagation. Specifically, the loss will affect the pruning probabilities $\boldsymbol{\pi}$ indirectly and then help to learn the prediction module. To make the attention mask plausible, we have a ratio loss that enforces the learned decision close to the pre-defined pruning ratio $\rho$ (Eq. 15). The effect of ratio loss can be found in the supplementary material.
>
> **Q5: About the zero token issue**
>
> **[Reply]** Actually, that is how we do during the inference, where uninformative tokens are directly discarded and the remaining tokens are made compact. However, if we do the same during training, the gradients cannot be back-propagated to the prediction modules, which makes the prediction modules tend to produce random pruning strategies. In our case, the discarded tokens must be involved in the computation graph but cannot have any influence on the remaining tokens. Therefore, we devise the attention masking strategy to resolve the above problems during training.

---

### Official Review · Reviewer_s33y · 2021-07-13

**Rating:** 7
**Confidence:** 3

**Summary:**

This paper works on accelerating the inference of ViT models by dynamically dropping tokens. This is realized by learning a light-weighted importance mask prediction between ViT blocks. The mask prediction module contains a few MLP layers and runs in $O(N)$ time instead of $O(N^2)$ in the self-attention. The authors also identified and resolved a few technical challenges in practice: 1. differentiating into the hard sampling; 2. batching after pruning. Experiments show the proposed method improves throughput by 43% ~ 54% with 0.2 ~ 0.5 accuracy drop.

**Limitations And Societal Impact:**

Yes

**Main Review:**

### strengths

- The overall results of improving throughput by 43% ~ 54% with 0.2 ~ 0.5 accuracy drop is favorable, and is likely to be used in practice in future ViT-based models.

 - The technical contribution of using MLPs on each individual token together with a global pooling of tokens to predict importance score makes sense. This is more efficient than the straightforward baseline of using the attention mask as the pruning weight (it will be great if such a baseline comparison is provided in the paper).

 - I like the model scaling figure (Fig. 4). It shows the proposed method works under large backbones.

 - Code is provided in the supplement.

### Weaknesses

 - While the overall throughput improvements are convincing, however I didn't understand where are the speed improvements from. In L143, the authors say in practice they don't actually remove the tokens due to batching. If this is the case, should the actually FLOPs keep unchanged? Is it the same for both training and testing? It seems that testing is also batched (Table. 1 caption). Please make sure to clarify this in the rebuttal.

 - The paper does not compare to other efficient ViT baselines. I know there are no neighbors directly close to this work, but some straightforward counterparts I have in mind are the O(n log n) transformers, or just directly pruning using the attention weight (e.g., the cross-attention weight of the class token). I will consider increasing my rating if such experiments are provided in the rebuttal and the results are interesting.

 - I don't quite like the distillation loss and KL loss. First, why do we need both distillation loss and KL loss? They seem pretty similar. It would be better if the author can provide ablations of both losses. Note this is NOT a required experiment in the rebuttal.

 - The distillation and Eq 9-11 seem to make the training much slower. Please report the training time (e.g., with/ without distillation) for reference. Much slower training time will NOT decrease my rating.

 - I don't get the problem of `the zeroed tokens will still influence other tokens through the Softmax operation` in L146. Would weight the zeroed token by negative infinite a simple solution, instead of using EQ 9-11?

 - The term `structural downsampling` in Fig. 1 is a bit confusing. From the context, it seems this just refers to pooling. `Structural downsampling` causes confusion to the term `structural pruning` in the network sliming literature, which refers to filtering the weights.

  - Should $D$ in Eq. 2 be $\hat{\bf D}$?

### Rating

  Overall this is a good paper with favorable experimental results, novel technical contribution, and good application value. Most of my negative comments are about writing, and I hope the authors can clarify them in the rebuttal.


**Time Spent Reviewing:**

3

---

> ### Author Response · Authors · 2021-08-10
> **Response to Reviewer s33y**
>
> We sincerely thank the reviewer for the positive comments on our work! We address the questions and clarify the issues accordingly as described below.
>
> **Q1: Token sparsification during inference**
>
> **[Reply]** Sorry for the confusion. During inference, we directly **drop** the tokens with less keeping probabilities as described in L175-L180. Therefore, the number of tokens gradually decreases during the pruning procedure and the computational costs are reduced. Besides, to make our method hardware friendly, we select a fixed number of remaining tokens during inference and only dynamically decide the locations of the remaining tokens as mentioned in Line 175. Thanks to the nature of ViT, removing tokens will not affect the computation of self-attention and MLP. Table 1 also shows our method can achieve significant throughput improvement on GPU with batch size > 1. We have also tried to dynamically decide the number of tokens during inference in our early experiments, but the improvement in accuracy is not significant while making the model hard to parallel. Therefore, we choose to fix the number of remaining tokens during inference in our final models.
>
>
> **Q2: Comparisons with other baselines**
>
> **[Reply]** Thanks for your advice. We conducted experiments by replacing the Attention module in DeiT with its efficient alternatives proposed by Performer [a] ($\mathcal{O}(n \log n)$) and Linformer [b] ($\mathcal{O}(n)$). We set both the number of (random orthogonal) features $m$ in Performer and the projected dimension $k$ as 64 to make the two variants have similar FLOPs to DeiT. The results are listed as follows:
>
> | model | GFLOPs | ImageNet Acc. (%)|
> |-----|----|------|
> | DeiT-S   |  4.6 | 79.8 |
> | Performer   |  4.5 | 54.8 |
> | Linformer   |  4.5 | 77.8  |
> | DynamicViT-S-0.9  |  4.0 | 79.8 |
>
> In our experiment, we have to use a relatively small $m$ in Performer to control the FLOPs, which might make the approximation of attention in Performer inaccurate and further lead to the worse performance of Performer. Besides, since we consider a relatively small number of tokens in vision problems (e.g., 14x14 tokens in DeiT), linear layers in the self-attention sub-layer and FFN contribute the most computations in vision transformers. Therefore, efficient attention will not significantly accelerate vision transformers. Instead, by removing redundant tokens, our method can simultaneously reduce the computation from both linear layers and the self-attention operation.
>
> **Q3: Ablations of distillation loss and KL loss**
>
> **[Reply]** Thank you for pointing out this. We have included some ablation in the supplementary and we conducted several extra experiments to prove the effectiveness of both the KL loss and the distillation loss:
>
> | Backbone Model | DeiT-S | LVViT-S |
> |------|------|-------|
> |DynamicViT | 79.32 (-0.48) | 83.0 (-0.3) |
> |w/o distill loss (Eq.13) | 79.30 (-0.50) | 82.72 (-0.58)|
> |w/o KL loss  (Eq.14) | 79.24 (-0.56) | 82.91 (-0.39) |
> |w/o distill loss, w/o KL loss|  79.17 (-0.63) | 82.52 (-0.78)|
>
> We see our method also works well without knowledge distillation losses. However, these two terms can consistently further boost performance for various models.
>
> **Q4: About the training time**
>
> **[Reply]** The training time of different configurations are listed as below:
>
> |model configuration|training time per epoch|
> |------|------|
> |DeiT baseline | 8m 51s |
> | +prediction module | 11m 19s |
> | +attention masking | 11m 36s  |
> | +distillation | 13m 36s |
>
> We find the attention masking (Eq. 9-11) won’t affect the training time a lot and introducing the distillation will bring 2 minutes extra training time per epoch.
>
> **Q5: The reason to use attention masking (Eq. 9-11)**
>
> **[Reply]**  We agree that filling the pruned tokens with negative infinity is equivalent to our attention masking. In our experiments, we find that introducing infinity will make the training unstable thus we devise an attention masking strategy to overcome this issue.
>
> **Q6: About the term structural downsampling**
>
> **[Reply]** Sorry for the confusion. By ```structural downsampling``` we aim to highlight the difference between the pooling in CNN and the token sparsification in DynamicViT. Maybe we can decide how to choose a proper term in further discussion.
>
> **Q7: About the typo**
>
> **[Reply]** Sorry for the typo. It should be $\hat{\mathbf{D}}$ which indicates the decision up to the current block. We will fix this typo in the revised version.
>
> [a] Choromanski, Krzysztof, Valerii Likhosherstov, David Dohan, Xingyou Song, Andreea Gane, Tamas Sarlos, Peter Hawkins et al. "Rethinking attention with performers." ICLR 2021.
>
> [b] Wang, Sinong, Belinda Z. Li, Madian Khabsa, Han Fang, and Hao Ma. "Linformer: Self-attention with linear complexity." arXiv preprint arXiv:2006.04768 (2020).

---

> > ### Comment · Reviewer_s33y · 2021-08-23
> > **After rebuttal**
> >
> > Thank you for your response. My main confusion (Q1) is resolved in the rebuttal. Also thank you for satisfying my curiosity about Linformer and Performer. The results are interesting but are not exactly clear given the limited content (and time). E.g., it's unclear why Performer is much worse than Linformer. The rest responses are satisfying. I am happy that the slow distillation loss only brings marginal improvements. Overall, I am more supportive of this paper. If I can, I would increase my rating to 7.5.

---

### Official Review · Reviewer_Gaez · 2021-07-15

**Rating:** 6
**Confidence:** 4

**Summary:**

The paper introduces a dynamic network to achieve efficient inference for vision transformers. Specifically, the proposed framework adopts a multi-stage architecture. In each stage, a learnable mask with the Gumbel-softmax training strategy is generated to remove some tokens for efficiency. The distillation loss and fine-tuning process are used for higher performance.

**Limitations And Societal Impact:**

Only partially, more discussion is needed.

**Main Review:**

Strength:
+ Albeit the dynamic mechanism has been widely explored on Transformers in NLP tasks and CNN networks in Image classification tasks, introducing dynamic token sparsiﬁcation for Vision Transformers is new and interesting.
+ The visualization and empirical analysis are clear and make sense.
+ The paper is generally well written.

Weakness:
- The experimental comparison is not that fair. The proposed framework uses additional distillation losses on features and predictions but the baseline methods do not. According to the observations in DeiT[21], the distillation procedure is essential for classification accuracy. Although some ablation studies are provided in the supplementary material, the performance of the baseline framework with the distillation procedure has not been revealed.
- Compared with the baseline methods, the DynamicViT adopts an additional fine-tuning procedure with 30 epochs. The fine-tuning procedure with distillation could significantly improve the performance of Vision Transformers, but the paper has not reported it.
- Can the DynamicViT be trained in an end-to-end manner? How about the performance if it is trained from scratch?

After Rebuttal:
The response addressed my concerns. I lean to vote for acceptance.

**Time Spent Reviewing:**

2

---

> ### Author Response · Authors · 2021-08-10
> **Response to Reviewer Gaez**
>
> Thanks for pointing out your concerns. We would like first to highlight the position of this paper here: we aim to design an effective method to perform token pruning rather than designing new transformer architectures. In the area of network pruning, it is a common practice to start from a pre-trained model and reduce the computational complexity with a minor performance drop.
>
> **Q1: About the distillation loss**
>
> **[Reply]** The distillation loss in our paper is different from the one used in DeiT.  The distillation loss in DeiT involves a more powerful CNN teacher model (RegNetY). However, in our paper, the distillation loss is calculated between the pruned model (our DynamicViT) and the pre-trained baseline model with the same architecture and initial state, which means we do not use any information other than the base model. We also note that knowledge distillation from the original model is also widely used in network pruning [a].
>
> **Q2: The fine-tuning procedure**
>
> **[Reply]** Related to the above, we note that the fine-tuning procedure requires no stronger teacher like RegNetY but only the base model itself. We believe that the performance of normal Vision Transformers cannot be improved via distillation if no stronger teacher is available. Besides, as reported in [b], 100 epochs longer training of a vision transformer can only bring 0.2% accuracy improvement on ImageNet.
>
>
> **Q3: The reason to use pre-trained models**
>
> **[Reply]** Our method is similar to the retraining [c] methods in the pruning literature, where fine-tuning is required to recover the accuracy drop from pruning. We choose to use a pre-trained model because our pruning strategy is highly related to the intrinsic sparsity in a well-trained vision transformer model. In other words, a pre-trained model can help the prediction module produce a better pruning strategy and select the most informative tokens.
>
> [a] Movement pruning: Adaptive sparsity by fine-tuning, NeurIPS 2020.
>
> [b] Going deeper with Image Transformers
>
> [b] Learning both weights and connections for efficient neural networks, NeurIPS 2015.

---

### Official Review · Reviewer_b7CH · 2021-07-19

**Rating:** 6
**Confidence:** 4

**Summary:**

Based on the observation that the final prediction in ViT is only decided by the most informative tokens, the author proposes DynamicViT which adopts dynamic token sparsification framework to prune redundant tokens progressively and hierarchically dependent on the input. To optimize the pruning in an end-to-end and differentiable way, the author proposes the attention masking strategy. The hierarchically pruning largely reduces the FLOPs and increases the throughout with the price of a very negligible performance drop for ViT, which demonstrates a very good complexity-accuracy trade-off achieved by the proposed DynamicViT.

**Ethics Review Area:**

["I don’t know"]

**Main Review:**

Strengths:
1) The paper is generally well-written and easy to follow. The code is also provided for reproducibility.
2) The DynamicViT model is inspired by a very interesting observation and the author conducts extensive experiments to confirm the correctness of the assumption.
3) The proposed attention masking strategy enables the DynamicViT module to be trained in a differentiable end-to-end manner.
4) DynamicViT achieves a very promising complexity-accuracy trade-off compared to CNNs and ViT approaches.

Weaknesses:
1) The hierarchically pruning of DynamicViT works well for the classification tasks. But it seems hard for DynamicViT to extend to object detection & semantic segmentation tasks, which limits the generalization and application scenarios of the proposed approach.

**Time Spent Reviewing:**

3 hours

---

> ### Author Response · Authors · 2021-08-10
> **Response to Reviewer b7CH**
>
>
> We sincerely thank the reviewer for the positive comments on our work! We address the questions and clarify the issues accordingly as described below.
>
> **Q1: Extension to other downstream tasks**
>
> **[Reply]** We think our framework can be extended to various downstream tasks like detection and segmentation with some modifications. One possible solution is recovering the deep representation for the pruned tokens from the remaining tokens and the features from the shallow layers via an additional reconstruction model. The reconstruction model can be jointly optimized by using the reconstruction loss and the objectives of the downstream tasks.  We agree that this is a promising direction to further improve our work. We would like to leave this as our future work.

---

### Official Review · Reviewer_NNmU · 2021-07-19

**Rating:** 6
**Confidence:** 5

**Summary:**

The intuition of this paper is that attention is sparse in vision transformers. To this end, this paper proposes a pruning method to estimate the importance score of each token before self-attention layer. An attention masking strategy is proposed to differentiably prune a token by blocking its interactions with other tokens. Experiments on DeiT and LV-ViT demonstrate the effectiveness of the proposed method.

**Limitations And Societal Impact:**

see above

**Main Review:**

No ablation studies on the loss functions

Although it's a pruning strategy on the DeiT and LV-ViT, the proposed method is act as a downsampling operation (L120 use C′ = C/2 in our implementation).
However, the baseline is only set to the model doesn't use the pruning method, so baseline comparison with the convolution based downsampling architecture is needed. Comparing with those method in Table 2 (eg PVT) is not enough. The authors need to design fair experiment setting to show the pruning downsampling is better than the convolution downsampling.

**Time Spent Reviewing:**

4

---

> ### Author Response · Authors · 2021-08-10
> **Response to Reviewer NNmU**
>
> We would like to thank the reviewer for the careful reading and valuable comments. We address the questions and clarify the issues accordingly as described below.
>
> **Q1: Ablation studies on the loss functions**
>
> **[Reply]** We have included some of the ablation studies on the loss functions in the supplementary material (Appendix B). We provide a more detailed ablation study here. We see the improvement brought by the two KD losses is not very significant, but it can consistently further boost the performance of various models.
>
>
> | Backbone Model | DeiT-S | LVViT-S |
> |------|------|-------|
> |DynamicViT | 79.32 (-0.48) | 83.0 (-0.3) |
> |w/o distill loss (Eq.13) | 79.30 (-0.50) | 82.72 (-0.58)|
> |w/o KL loss  (Eq.14) | 79.24 (-0.56) | 82.91 (-0.39) |
> |w/o distill loss, w/o KL loss|  79.17 (-0.63) | 82.52 (-0.78)|
>
> **Q2: Comparison with convolution downsampling**
>
> **[Reply]** To fairly compare with convolution downsampling, we conduct another experiment where we use a convolution operation with stride as 2 after the sixth block of the baseline DeiT-S model, which has similar FLOPs and network architecture to our DynamicViT model. The model is trained from scratch using the identical training strategy with DeiT. The accuracy on ImageNet is 77.0% while our DynamicViT can achieve 79.3%. This indicates our method is better than convolution downsampling under a fair setting. Besides, we have also compared a baseline model with structural downsampling in Table 3, where we add an average pooling layer to the sixth block of a pre-trained DeiT-S model and finetune the model using our proposed distillation losses for 30 epochs (we didn’t add a convolutional downsampling layer since new parameters will spoil the well-pretrained model). We see our dynamic sparsification is significantly better than conventional structural downsampling (1.1% lower than DynamicViT on ImageNet). These results show DynamicViT is more efficient than CNN-style downsampling in both settings.  We will add more discussions and results on this issue in the revised paper.

---

> > ### Comment · Reviewer_NNmU · 2021-08-11
> > **follow-up**
> >
> > Thank you for your response. I am pleased with the clarification and additional experiments.

---

### Official Review · Reviewer_z3xW · 2021-07-20

**Rating:** 7
**Confidence:** 4

**Summary:**

This paper proposes a lightweight prediction module to estimate the importance score of each token in the vision transformer. The module is added to different layers to prune redundant tokens hierarchically. To optimize the prediction module in an end-to-end manner, an attention masking strategy is proposed to differentiably prune a token by blocking its interactions with other tokens. Experiments show that the proposed method can significantly reduce the computational cost with a small drop in accuracy.

**Main Review:**

I think that it is an interesting idea to utilize the natural properties of the vision transformer for network pruning. Maybe it is the first paper in this topic. The experimental results are also pretty good. The writing is clear and professional. However, this paper is only evaluated for image classification. I’m wondering if the proposed pruning affects much for downstream vision tasks (e.g., object detection and semantic segmentation)?

**Time Spent Reviewing:**

Three hours

---

> ### Author Response · Authors · 2021-08-10
> **Response to Reviewer z3xW**
>
> We sincerely thank the reviewer for the positive comments on our work! Please see our below responses to the concerns.
>
> **Q1: Extension to other downstream tasks**
>
> **[Reply]** Thanks for your advice. We think our framework can be extended to various downstream tasks like detection and segmentation with some modifications. One possible solution is recovering the deep representation for the pruned tokens from the remaining tokens and the features from the shallow layers via an additional reconstruction model. The reconstruction model can be jointly optimized by using the reconstruction loss and the objectives of the downstream tasks.  We agree that this is a promising direction to further improve our work. We would like to leave this as our future work.

---

### Decision · Program_Chairs · 2021-09-28

**Decision:**

Accept (Poster)

**Comment:**

All the reviewers agree that the observations made by this submission are interesting and the proposed dynamic token sparsification method is novel and effective. Some concerns on the experiment details are raised by the reviewers but the authors address them well in the response. A clear acceptance.

**Consistency Experiment:**

NeurIPS has a long history of experimentation. In 2014, NeurIPS ran an experiment in which 10% of submissions were reviewed by two independent committees to quantify the randomness in the review process. This year, we repeated a variant of this experiment to see how the quality of the review process has changed over time.  This paper was part of the experiment and was therefore assigned to two committees (consisting of reviewers, an Area Chair, and a Senior Area Chair) that reached independent decisions.  If both committees made the same recommendation, this recommendation was followed. If a single committee recommended acceptance, the paper was accepted (with the exception of a few cases in which the other committee identified what we considered a fatal flaw, e.g., an error in a key result).

Both committees reached the same decision: **Accept (Poster)**

The other committee assigned to the paper recommended **Accept (Poster)**.  You can find the other set of reviews, along with any follow up discussion with the authors here:
https://openreview.net/forum?id=jB0Nlbwlybm